# Melatonin Ameliorates Post-Stroke Cognitive Impairment in Mice by Inhibiting Excessive Mitophagy

**DOI:** 10.3390/cells13100872

**Published:** 2024-05-18

**Authors:** Yan Shi, Qian Fang, Yue Hu, Zhaoyu Mi, Shuting Luo, Yaoxue Gan, Shishan Yuan

**Affiliations:** 1Key Laboratory of Study and Discovery of Small Targeted Molecules of Hunan Province, School of Medicine, Hunan Normal University, Changsha 410006, China; shiy@hunnu.edu.cn (Y.S.); luoshuting@hunnu.edu.cn (S.L.); 2Department of Medical Laboratory, School of Medicine, Hunan Normal University, Changsha 410006, China; fangqian@hunnu.edu.cn (Q.F.); 202220193561@hunnu.edu.cn (Y.H.); zoeym0323@hunnu.edu.cn (Z.M.); kwx0130@hunnu.edu.cn (Y.G.); 3Engineering Research Center of Reproduction and Translational Medicine of Hunan Province, School of Medicine, Hunan Normal University, Changsha 410013, China

**Keywords:** post-stroke cognitive impairment, melatonin, mitophagy, synaptic plasticity, neuroprotective

## Abstract

Post-stroke cognitive impairment (PSCI) remains the most common consequence of ischemic stroke. In this study, we aimed to investigate the role and mechanisms of melatonin (MT) in improving cognitive dysfunction in stroke mice. We used CoCl_2_-induced hypoxia-injured SH-SY5Y cells as a cellular model of stroke and photothrombotic-induced ischemic stroke mice as an animal model. We found that the stroke-induced upregulation of mitophagy, apoptosis, and neuronal synaptic plasticity was impaired both in vivo and in vitro. The results of the novel object recognition test and Y-maze showed significant cognitive deficits in the stroke mice, and Nissl staining showed a loss of neurons in the stroke mice. In contrast, MT inhibited excessive mitophagy both in vivo and in vitro and decreased the levels of mitophagy proteins PINK1 and Parkin, and immunofluorescence staining showed reduced co-localization of Tom20 and LC3. A significant inhibition of mitophagy levels could be directly observed under transmission electron microscopy. Furthermore, behavioral experiments and Nissl staining showed that MT ameliorated cognitive deficits and reduced neuronal loss in mice following a stroke. Our results demonstrated that MT inhibits excessive mitophagy and improves PSCI. These findings highlight the potential of MT as a preventive drug for PSCI, offering promising therapeutic implications.

## 1. Introduction

A stroke is an acute cerebrovascular event characterized by clinical signs of focal or global impairment of cerebral function and is a leading cause of death and disability worldwide [1]. Ischemic stroke, which accounts for approximately 71% of all strokes, is caused by an insufficient blood supply or an embolus in the cerebral vasculature [2]. The sudden interruption of cerebral blood flow, along with the deprivation of oxygen and glucose supply to the brain tissue, leads to the disruption of cellular homeostasis and subsequent ischemic neuronal injury [3]. Post-stroke cognitive impairment (PSCI) remains a severe and typical consequence of ischemic stroke and is a broad concept that encompasses the full spectrum, from mild cognitive impairment to dementia [4]. Global cognitive deficits are detectable in approximately 44% of stroke patients admitted to hospital, according to data harmonized from thirteen studies based in eight countries [5]. Previous estimates have suggested that patients with PSCI have a higher chance of death than those without PSCI, and their lifestyles are significantly affected [6]. Furthermore, PSCI affects survivors of all ages, with evidence that even 50% of young stroke patients (under 50 years of age) exhibit poor cognitive performance that persists for up to 11 years after the original brain injury, leading to increased stroke mortality [7]. Although extensive efforts have been devoted to studies on strokes, research on improving cognitive function after an ischemic stroke is lacking. Therefore, an increased understanding of the underlying pathogenesis of PSCI and the development of targets to prevent it are of critical importance in current stroke research and therapies.

Mitochondria, the powerhouse of cells, are critical for cellular energy homeostasis and neurological improvement following an ischemic stroke. During an ischemic stroke, inadequate delivery of oxygen and nutrients can immediately lead to mitochondrial dysfunction [8]. In response to stress conditions, impaired mitochondria trigger the initiation and amplification of caspase-dependent apoptotic cascades and alter synaptic connections, thereby exacerbating cell death and contributing to secondary cerebral damage [9,10]. Mitophagy is a selective process that eliminates damaged mitochondria and plays an important role in controlling mitochondrial quality and cellular homeostasis [11]. Basal mitophagy is indispensable for cell survival, while excessive mitophagy leads to autophagy-dependent neuron death [12]. Moreover, excessive mitophagy may lead to cognitive impairment. It has been shown that abnormally excessive mitophagy leads to a significant accumulation of ubiquitinated proteins, which results in impaired synaptic plasticity and ultimately leads to cognitive dysfunction in mice [13]. Therefore, focusing on mitophagy may be a promising approach for PSCI therapy.

N-acetyl-5-methoxytryptamine (melatonin, MT), an endogenous hormone produced by the pineal gland, has a wide range of regulatory and protective effects, such as neuroprotection, synchronization of circadian rhythms, regulation of energy metabolism, and protection against oxidative stress [14]. MT ameliorates cognitive function in mice following a traumatic brain injury by inhibiting astrocyte reactivation [15]. In addition, another study showed that MT could inhibit neuroinflammation in the hippocampus by suppressing the TLR4/MyD88/NFκB signaling pathway, thereby improving methamphetamine-induced cognitive impairment [16]. However, whether MT improves PSCI and its specific mechanisms are not well understood.

The hippocampus is an important component of the limbic system that is involved in learning and memory consolidation [17]. It has been shown that the volume of the ipsilateral hippocampus markedly decreases in patients with unilateral middle cerebral artery occlusion, and cognition is severely impaired when the hippocampal shrinkage ratios exceed 20% [18]. Delattre et al. demonstrated that abnormally low neuronal soma surface areas in the CA1, CA3, and CA4 regions of the ischemic rat hippocampus led to inward deformation of the hippocampus, ultimately leading to cognitive deficits [19]. Therefore, it is particularly important to explore the pathological mechanisms underlying hippocampal injury in PSCI to develop new methods for prevention and treatment.

While previous research has laid a foundation for understanding the broad implications of PSCI and the role of mitochondrial dysfunction in neurological recovery, our understanding of the specific molecular mechanisms underlying hippocampal injury in PSCI remains unclear. Furthermore, although studies have explored the therapeutic potential of MT in various neurological conditions, its efficacy and specific mechanisms in ameliorating PSCI remain inadequately understood. In this study, we used the human neuroblastoma cell line SH-SY5Y to construct a hypoxic injury cell model and photochemically-induced thrombosis to develop a mouse model of stroke. We sought to explore the changes in molecular mechanisms in the hippocampus and the effects of MT on cognitive function in stroke mice. Additionally, we aimed to investigate whether mitophagy plays a role in this process. Our findings will provide an improved understanding of PSCI pathogenesis and help identify potential therapeutic targets for intervention.

## 2. Materials and Methods

### 2.1. Animals

Eight-week-old male C57/BL6J mice were obtained from Hunan Slake Jingda Laboratory Animal Co., Ltd. (Changsha, China). The mice were randomly housed in cages (5 mice per cage) under a 12 h light/dark cycle, constant temperature (22 ± 2 °C) and humidity (50 ± 10%), and free access to water and food ad libitum. Mice had a 1-week acclimatization period before the following experiments. All animal experiments were approved by the Animal Care and Use Committee of the Hunan Normal University, and all measures were taken to reduce animal numbers and suffering.

### 2.2. Photothrombotic (PT) Infarction Model

Mice were anesthetized by inhaling 2% isoflurane and being injected with Rose Bengal (10 mg/mL in 0.9% saline; Sigma-Aldrich, St. Louis, MO, USA) on the other side of the peritoneum to induce photothrombotic stroke. Subsequently, the mice’s head hair was shaved off and placed in a brain stereotaxic apparatus (David Kopf instrument, Tujunga, CA, USA). After sterilizing the scalp with iodophor, the skull was exposed, the periosteum was removed by wiping with hydrogen peroxide, and the cranial bones were kept dry. Five minutes after the injection of Rose Bengal, the skull was illuminated for 20 min using a cold light source positioned 2 mm away from the right lateral side of the bregma, with the light source as close as possible to the skull.

### 2.3. Experimental Groups and Drug Treatment

In experiment 1, all experimental animals were randomly divided into two groups: sham and stroke (n = 11 in each group). The stroke group underwent photothrombotic modeling as described above, whereas the sham group received an intraperitoneal injection of Rose Bengal without illumination. Behavioral experiments began on the 22nd day after modeling.

In experiment 2, all experimental animals were randomly divided into three groups: sham, stroke, and stroke + MT, with n = 7 for each group. Intraperitoneal injection of MT (20 mg/kg in 0.9% saline; Sigma-Aldrich, St. Louis, MO, USA) was administered on the first day after the end of the modeling, while the sham and stroke groups were administered an equal volume of saline for 21 days. Behavioral experiments began on the 22nd day.

### 2.4. 2,3,5-Triphenyl-tetrazolium Chloride (TTC) Staining

To determine whether the photothrombotic infarction model was successfully constructed, mice were anesthetized and processed by decapitation 24 h after stroke (n = 4 in each group). The brains were quickly removed and immediately placed in a −20 °C refrigerator for 30 min. After being cut into thick coronal sections, the brain slices were incubated in a 2% TTC (MeilunBio, Dalian, China) solution at 37 °C in the dark for 30 min. Subsequently, photographs were captured to assess the extent of infarction.

### 2.5. Assessment of Neurological Severity Score

Neurological deficits in the mice were measured using the modified neurological severity score (mNSS), which assesses motor function, sensory response, balance, and reflex/abnormal movements comprehensively. The mNSS scores range from 0 to 18, with higher scores indicating more severe neurological damage. The scores for the mice in each group were obtained 24 h after modeling.

### 2.6. Novel Object Recognition Test (ORT)

The device consisted of a black-walled square box measuring 40 × 40 × 40 cm. On the first day, mice were placed in an open-field box and allowed to move freely for 10 min. On the second day, two identical objects were placed diagonally opposite to each other in the box, and the mice were allowed to explore freely for 5 min. Active exploration was defined as sniffing or approaching objects less than 2 cm in size. On the last day, one familiar object was replaced with a novel object, and it was switched to the other two diagonal corners to eliminate the effect of the mice’s positional preference on the experiment. A video camera was used to record the activity of the mice as they explored either the new or old object for 5 min. To avoid coercion while exploring the objects, we placed the mice with their backs toward the objects and equidistant from them. The videos were analyzed at the end of the experiment using the SuperMaze video 2.0 analysis software (Shanghai XinRuan Information Technology Co., Ltd., Shanghai, China), and the relative exploration time was expressed as a discrimination index calculated as new object exploration time/(new object exploration time + old object exploration time) × 100%.

### 2.7. Y Maze

The three arms of the Y maze were randomly set as the start arm, novel arm, and other arm, with the individual arms at an angle of 120°. The maze was lined with wood shavings. After each training or testing session, the wood shavings in each arm were mixed, and the maze was wiped with alcohol to prevent interference from residual animal odors. Different geometric shapes were labeled inside each arm as visual markers. During the training phase, mice were placed at the distal end of the starting arm facing the center of the maze for 10 min, while the novel arm was blocked with a spacer. One h after the end of the training period, the novel arm spacer was withdrawn, and the mice were placed in the starting arm again for 5 min. The time the mice spent in each arm during the 5 min was recorded on video. The percentage of time spent in the novel arm (%) was calculated using the following formula: (novel arm exploration time/total exploration time) ×100.

### 2.8. Nissl Staining

Mice were perfused with 4% paraformaldehyde, and their brains were collected. Once the brain sank, it was transferred sequentially to 10% and 30% sucrose solution for gradient dehydration and then cut into 30 μm slices using a frozen sectioning machine (Leica, Wetzlar, Germany). The hippocampal region of the mouse brain slice was stained using a Nissl staining kit (Solarbio, Beijing, China), according to the manufacturer’s instructions. The images were statistically analyzed using the ImageJ 1.53 software.

### 2.9. Assessment of Relative mtDNA Copy Number

Mitochondrial coding gene *mt-Atp6* represents mitochondrial content, ribosome-encoded gene *Rpl13* represents cellular content, and the ratio of *mt-Atp6*:*Rpl13* represents the relative content of mitochondria in cell. DNA from hippocampal tissue was extracted with a DNA Extraction Kit (Solarbio, Beijing, China) according to the manufacturer’s instruction. The relative mtDNA copy number was measured by qPCR. The qPCR was conducted as follows: 10 min at 95 °C, 15 s at 58 °C, and 20 s at 72 °C. Primer sequences were as follows: *mt-Atp6* (Forward: 5′-ATTACGGCTCCTGCTCATA-3′; Reverse: 5′-TGGCTCAACCAACCTTCTA-3′) and *Rpl13* (Forward: 5′-TGATTGGCGTTTGAGATTGGC-3′; Reverse: 5′-AATCCTTGTGGAAGTGGGGC-3′). Relative DNA levels were calculated using the 2^−ΔΔCt^ method.

### 2.10. Cell Culture and Treatment

Human SH-SY5Y cells were routinely cultured in Dulbecco’s modified Eagle medium (Gibco, NY, USA) with 10% fetal bovine serum (Biological Industries, Kibbutz Beit Haemek, Israeli) and 1% antibiotics (Penicillin and Streptomycin; Biosharp, Hefei, China) at 37 °C in a humidified atmosphere containing 5% CO_2_.

### 2.11. Cell Counting Kit-8 (CCK-8) Assay

Briefly, SH-SY5Y cells were seeded in 96-well plates at a density of 5 × 10^3^ cells/well, and CCK-8 (Genview, Beijing, China) solution was added to each well after 24 h of treatment. The cells were incubated at 37 °C for 1 h, and the optical density at 450 nm was measured using a microplate reader (Heales, Shenzhen, China).

### 2.12. TUNEL Staining

The TUNEL assay was performed using the one-step TUNEL apoptosis assay kit (MeilunBio, Dalian, China), according to the manufacturer’s instructions. After TUNEL staining, the slices were blocked with an anti-fluorescence quenching blocker containing DAPI (Beyotime, Shanghai, China). TUNEL-positive and DAPI-positive cells were visualized by fluorescence microscopy.

### 2.13. Immunofluorescence Staining

After treatment with CoCl_2_ and MT, SH-SY5Y cells were fixed with 4% paraformaldehyde (Biosharp, Hefei, China) for 15 min and incubated with 0.2% Triton X-100 (Yeasen, Shanghai, China) for 10 min at 25 °C. Next, SH-SY5Y cells were blocked with 5% bovine serum albumin (BSA) for 30 min, and then incubated overnight at 4 °C with translocase of the outer mitochondrial membrane 20 (Tom20; Proteintech, Wuhan, China; 1:400) and LC3 (Proteintech, Wuhan, China; 1:400) antibodies. Fluorescence images were obtained using a fluorescence microscope (Thermo Fisher Scientific, Waltham, MA, USA).

### 2.14. Transmission Electron Microscopy (TEM)

The SH-SY5Y cells were recovered by centrifugal separation and washed with PBS, and the cell precipitates were suspended and fixed with 2% glutaraldehyde. After sectioning and uranium–lead double staining, the samples were observed and photographed using a TEM system (Hitachi, Tokyo, Japan).

### 2.15. Western Blot Analysis

The protein levels of cells and hippocampal tissues were determined using cold lysis buffer (PMSF:RIPA = 1:100), and protein concentrations were measured according to the manufacturer’s instructions using a BCA kit (Solarbio, Beijing, China). Proteins were separated using SDS-PAGE, transferred onto polyvinylidene difluoride membranes, and blocked with 5% BSA at room temperature for 1 h. The membranes were then incubated overnight at 4 °C with the following primary antibodies: anti-Bax (Beyotime, Shanghai, China; 1:1000), anti-Bcl2 (Beyotime, Shanghai, China; 1:500), anti-Caspase-3 (Beyotime, Shanghai, China; 1:1000), anti-cleaved Caspase-3 (Cell Signaling Technology, Danvers, MA, USA; 1:1000), anti-β-actin (ABclonal, Wuhan, China; 1:50,000), anti-dynamin-related protein 1 (Drp1; Proteintech, Wuhan, China; 1:2000), anti-Tom20 (Proteintech, Wuhan, China; 1:10,000), anti-PINK1 (Proteintech, Wuhan, China; 1:1000), anti-Parkin (Proteintech, Wuhan, China; 1:1000), anti-LC3 (Abcam, Cambridge, MA, USA; 1:1000), anti-P62 (Huabio, Hangzhou, China; 1:500), anti-PSD-95 (Beyotime, Shanghai, China; 1:1000), anti-Synaptophysin (Beyotime, Shanghai, China; 1:500), and anti-α-tubulin (Proteintech, Wuhan, China; 1:20,000). Then, the membranes were incubated with corresponding secondary antibodies for 1 h at room temperature. Finally, the membranes were scanned using an imaging system (Tanon-5200, Shanghai, China) and quantified using the ImageJ software.

### 2.16. Statistical Analysis

The data are presented as the mean ± standard error (SEM). Statistical analyses were performed using one-way ANOVA with the GraphPad Prism 5 statistical software. Statistical significance was set at *p* < 0.05.

## 3. Results

### 3.1. CoCl_2_ Impaired Neuronal Synaptic Plasticity and Upregulated Mitophagy Levels 

CoCl_2_ had toxic effects on the SH-SY5Y cells in a dose-dependent manner, with concentrations ranging from 50 μM to 800 μM (Figure 1A). Cell viability decreased significantly after treatment with 400 μM CoCl_2_, reducing to approximately 75% of the Con group. Thus, 400 μM CoCl_2_ was selected for subsequent experiments. The Western blotting results showed that CoCl_2_ significantly increased the levels of the pro-apoptotic proteins Bax and cleaved caspase-3 while decreasing the level of the anti-apoptotic protein Bcl-2 (Figure 1B,C). Additionally, the TUNEL staining results showed that the proportion of TUNEL-positive cells in the CoCl_2_ group was significantly higher than that in the control group (Figure 1D,E). These data indicated that treatment with 400 μM CoCl_2_ results in decreased cell viability and induced neuronal apoptosis. Furthermore, the levels of PSD-95 and synaptophysin significantly decreased, indicating impaired neuronal synaptic plasticity (Figure 1F,G).

To investigate whether mitophagy is involved in CoCl_2_-induced hypoxic injury, we examined the levels of autophagy proteins (LC3 and p62), mitochondrial proteins (Drp1 and Tom20), and mitophagy proteins (PINK1 and Parkin) using Western blotting. Compared with the Con neurons, the CoCl_2_ neurons showed elevated levels of LC3-II/LC3-I (a typical autophagic marker) and significantly decreased levels of p62, indicating the activation of autophagy. Moreover, the levels of Drp1, PINK1, and Parkin significantly increased, whereas the level of Tom20 decreased (Figure 1H,I). These results suggest that CoCl_2_ upregulates mitophagy.

### 3.2. The Stroke Mice Showed Cognitive Dysfunction and Neurologic Damage

TTC staining was performed 24 h after modeling, and the results showed that massive cerebral infarctions occurred in the stroke group, indicating successful stroke modeling (Figure 2A). The stroke mice showed high neurological deficit scores (Figure 2B), indicating impaired neurological function. The cognitive function of the mice was assessed using ORT and Y maze tests. The results showed that the stroke mice had a lower discrimination index and spent less time in the novel arm of the Y maze than the sham mice (Figure 2C,D). In addition, the stroke mice showed a reduced number of Nissl-positive neurons in the CA1, CA3, and DG subregions of the hippocampus and cortex (Figure 2E–I). These results revealed that cognitive dysfunction and neurological damage occurred in the stroke mice.

### 3.3. The Stroke Mice Had Upregulated Mitophagy Levels and Impaired Synaptic Plasticity

Based on the in vitro results, we further explored the underlying molecular mechanisms in the hippocampi of mice following a stroke. Compared with the sham mice, the stroke mice showed an increase in the levels of Bax and cleaved caspase-3 and a significant decrease in the levels of Bcl-2 (Figure 3A,B), suggesting that apoptosis had occurred. The levels of PSD-95 and synaptophysin were reduced (Figure 3C,D), indicating impaired synaptic plasticity. Increased levels of LC3-II/LC3-I, Drp1, PINK1, and parkin, as well as decreased levels of p62 and Tom20, were also observed (Figure 3E,F). In addition, the levels of mtDNA in the stroke mice were significantly reduced, which together suggested the upregulation of mitophagy in the stroke mice (Figure 3G).

### 3.4. MT Protected Neuronal Synaptic Plasticity and Downregulated Mitophagy Levels In Vitro

Based on our previous study, we used 100 μM MT to explore its protective effects [20]. The results showed that 100 μM MT significantly reversed the decrease in cell viability after CoCl_2_ treatment (Figure 4A). MT reversed the increase in Bax and cleaved caspase-3 levels and the decrease in Bcl-2 levels (Figure 4B,C), indicating that MT blocked apoptosis induced by CoCl_2_ treatment. The TUNEL staining also verified this observation, as MT significantly reduced the rate of neuronal apoptosis (Figure 4D,E). Next, we examined the effects of MT on neuronal synaptic plasticity. The Western blotting results showed that the decreases in PSD-95 and the synaptophysin levels were significantly reversed by MT (Figure 4F,G), suggesting a protective effect of MT on synaptic plasticity.

We further explored whether MT is involved in the regulation of mitophagy. Figure 4H,I clearly show that MT downregulated mitophagy, as demonstrated by decreased LC3-II/LC3-I, Drp1, PINK1, and Parkin levels and increased p62 and Tom20 levels. We used biochemical markers to examine the co-localization of LC3 (an autophagy marker) and Tom20 (a mitochondrial marker). The results showed that CoCl_2_ significantly promoted the co-localization of LC3 and Tom20 in neurons, whereas this co-localization was suppressed in CoCl_2_ + MT neurons (Figure 4J,K). In addition, we observed the fine structure of the mitochondria and changes in mitophagy using TEM. As shown in Figure 4L, the cell structure was clear, and the mitochondrial morphology was normal in the Con neurons. After CoCl_2_ treatment, multiple vacuoles appeared in the cytoplasm, the mitochondrial structure was disrupted, and widespread mitophagy (green arrow) was observed. In contrast, the CoCl_2_ + MT neurons showed a significant decrease in cell structure disruption, as well as a decrease in the number of vacuoles in the cytoplasm and a significant decrease in the amount of mitophagy. Notably, MT downregulated mitophagy levels and alleviated cellular damage caused by CoCl_2_ treatment.

### 3.5. MT Ameliorated Cognitive Defects and Reversed Neuron Loss in Stroke Mice

MT was administered intraperitoneally for 21 days after the end of modeling. Behavioral tests, Nissl staining, and Western blotting were begun on day 22 to assess the effects of MT on the neuronal and cognitive functions of stroke mice. Behavioral experiments showed that the stroke + MT mice had a higher discrimination index (Figure 5B) and spent a longer time in the Y maze novel arm than the stroke group (Figure 5C). In addition, the Nissl staining showed that the MT-injected mice had more Nissl-positive neurons and lower neuronal loss in the CA1, CA3, and DG subregions of the hippocampus and cortex than the stroke mice (Figure 5D–H). These results suggest that MT ameliorates cognitive deficits and reverses neuronal loss in stroke mice.

### 3.6. MT Ameliorated Synaptic Plasticity Impairments and Downregulated Mitophagy Levels in Stroke Mice

Western blotting showed that stroke + MT mice exhibited lower Bax and cleaved caspase 3 levels and higher Bcl-2 levels than the stroke group (Figure 6A,B), suggesting that MT inhibited apoptosis in mice. In addition, stroke + MT mice had lower levels of synaptophysin and PSD-95 than stroke mice (Figure 6C,D), indicating ameliorated synaptic plasticity impairment. MT significantly reversed the increase in LC3-II/LC3-I, Drp1, PINK1, and Parkin levels and decreased p62 and Tom20 levels, suggesting that MT could inhibit mitophagy (Figure 6E,F). In addition, the mtDNA levels were significantly increased in the stroke + MT mice (Figure 6G). These results indicate that MT ameliorated synaptic plasticity and downregulated mitophagy in stroke mice, which is consistent with the results from the in vitro experiments.

## 4. Discussion

Stroke is the leading cause of death and disability worldwide and imposes a heavy burden on family lives [21]. As a major complication of ischemic strokes, PSCI seriously affects the life of patients and increases the risk of disability, depression, and death [22]. However, research on the treatment of PSCI is limited. MT, a natural antioxidant, prevents the progression of mitochondrial-related diseases and neurological disorders [23,24]. However, whether MT also has a protective effect against PSCI and its underlying mechanisms remain to be explored. In this study, we demonstrated that MT reduced apoptosis and synaptic plasticity impairments and downregulated mitophagy both in vivo and in vitro, thereby ameliorating cognitive deficits and reducing neurological impairments after stroke.

CoCl_2_ can inhibit the formation of oxygenated hemoglobin by competing with divalent iron ion activity [25]; hence, it can be used to induce hypoxia while downregulating the concentration of serum in the medium to simulate a stroke at the cellular level [26]. Previous studies have shown that CoCl_2_ induces apoptosis [27], which is consistent with our experimental results. Synaptic plasticity refers to the modifiable strength of neuronal synaptic connections [28]. PSD-95 and synaptophysin are important synaptic proteins found in the neurons involved in the regulation of synapse formation, development, and plasticity [29,30]. Our results showed that CoCl_2_ downregulated synaptophysin and PSD-95 levels, suggesting impaired neuronal synaptic plasticity.

In vivo, we used the photothrombotic infarction model to advance our understanding of strokes, as the model induces a photochemical reaction in the brain that releases oxygen free radicals, ultimately leading to the interruption of blood flow as a result of platelet aggregation and alteration of the blood–brain barrier, which is consistent with stroke in humans [31]. Notably, several studies have confirmed that a PT-induced stroke can cause cognitive impairment [32,33,34]. We used two well-recognized behavioral paradigms, the NOR test and Y maze, to assess the effects of a stroke on cognitive function in mice. Both experiments were established as tests of learning and memory, utilizing the principle that mice have an innate tendency to explore new objects. Stroke mice showed a reduction in the exploration time for new objects (ORT) and novel arms (Y maze), suggesting impaired cognitive function. The hippocampus is the main brain region of the central nervous system that performs cognitive functions, and damage to the hippocampus frequently accompanies PSCI [35]. Our results showed that stroke mice exhibit increased apoptosis and impaired neuronal synaptic plasticity in the hippocampus.

An inadequate supply of oxygen and nutrients leads to the disruption of mitochondrial homeostasis and mitochondrial dysfunction and has been well-documented as a precursor to cell death [36]. Therefore, mitochondrial quality control by mitophagy is essential for metabolic homeostasis and cell survival [37]. Recent findings support the fact that the inhibition of RIPK3/AMPK-mediated mitophagy is beneficial for reducing neuronal apoptosis in mice following an ischemic stroke [38]. When mitochondria are damaged, PINK1 accumulates on the outer mitochondrial membrane and specifically recruits Parkin for translocation to the damaged mitochondria, thereby inducing mitophagy [39]. In this process, Drp1 acts synergistically with Parkin in mitochondrial biogenesis and degradation, thereby jointly promoting mitophagy [40]. The activation of mitophagy can promote the degradation of p62 and Tom20, thereby reducing the accumulation of damaged mitochondria.

However, the relationship between mitophagy and cognitive function remains controversial. It had been proposed that promoting mitophagy in AD mice could improve cognitive function by increasing the efficiency of microglia in clearing amyloid-β plaques and inhibiting the phosphorylation of tau protein [41]. In a mouse model of a stroke, the upregulation of mitophagy ameliorated cognitive deficits by inhibiting NLRP3 inflammasome activation [42]. However, excessive mitophagy can also lead to impaired cognitive function. Chronic cerebral hypoperfusion (CCH) can induce excessive activation of mitophagy, resulting in impaired cognition and memory abilities, whereas the inhibition of excessive mitophagy in CCH rats attenuates cognitive deficits and exerts neuroprotective effects [43]. Our study showed that CoCl_2_ induced excessive mitophagy, with increased co-localization of LC3 and Tom20 being observed under fluorescence microscopy and a large amount of mitophagy being observed in TEM. In addition, the levels of mitophagy-related proteins were upregulated in the hippocampi of the stroke mice, which was consistent with the in vitro results. Therefore, we believe that downregulating mitophagy in the hippocampus may be key to improving PSCI.

MT is a mitochondria-targeted antioxidant due to its amphiphilic nature making it easy to cross the mitochondrial membrane and be selectively taken up by the mitochondria [44]. There is growing evidence of its ability to regulate mitochondrial homeostasis, and it is widely recognized as having significant therapeutic potential in neurodegenerative diseases [45]. However, the role of MT in regulating mitophagy remains controversial. Studies have confirmed that MT exerts its protective effects on the nervous system by inducing mitophagy. In a neonatal hypoxic-ischemic encephalopathy model, intraperitoneal injection of MT (15 mg/kg) into pups 1 h before hypoxia induction upregulated the expression of NLRX1, which downregulated mTOR expression and promoted ATG7 and Beclin-1 expression to promote mitophagy and inhibit apoptosis [46]. In a model of glutamate-induced cytotoxicity in mouse hippocampal neurons, pretreatment with 10^−7^ mol/L MT for 2 h mediated mitophagy via the Beclin-1/Bcl-2 pathway, thus exerting antioxidative stress and neuroprotective effects [47]. In contrast, several studies have shown the inhibitory effect of MT on mitophagy. In the ropivacaine-induced PC12 and HT22 neurotoxicity cell models, pretreatment with 10 μM MT for 2 h reduced ropivacaine-induced apoptosis by inhibiting excessive mitophagy via the PINK1/Parkin pathway [48]. In the anoxia/reoxygenation (A/R) injury model, the administration of 150 μM MT at the onset of reoxygenation protected H9c2 cells from A/R injury by inhibiting excessive mitophagy [49]. The variation in the effects of MT on the regulation of mitophagy may be due to differences in experimental conditions and models. Our results showed that MT downregulates mitophagy in both cellular and animal models of strokes, which ameliorates neurological damage.

In conclusion, our study confirmed that MT inhibited excessive mitophagy, protected neuronal synaptic plasticity, and ameliorated PSCI. Our findings underscore MT’s potential as a preventive drug for PSCI, offering a promising therapeutic avenue. By elucidating MT’s mechanisms in modulating mitophagy and synaptic plasticity, we improve the understanding of PSCI’s pathophysiology and identify new intervention targets. MT’s neuroprotective effects highlight its significance in alleviating cognitive deficits and neurological impairments post-stroke. Moreover, as an approved and marketed mature drug, MT has been confirmed to have powerful neuroprotective effects in a large number of basic science studies, with high levels of safety and wide application prospects. Our further work aims to investigate the effects of MT on mitophagy in the ischemic cerebral cortex and other cognition-related brain regions, and to regulate the level of mitophagy to clarify its key role in PSCI.

## Figures and Tables

**Figure 1 cells-13-00872-f001:**
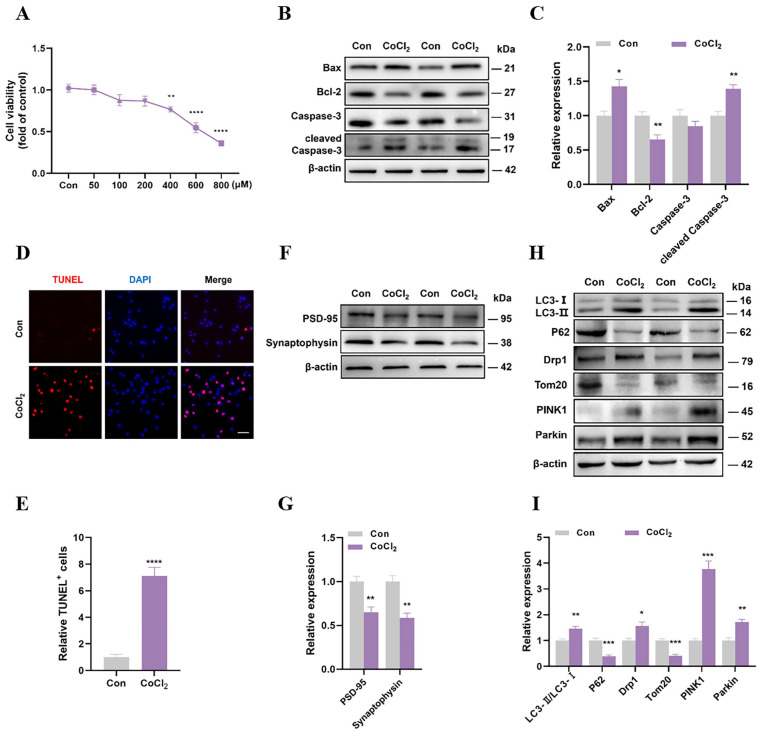
CoCl_2_ impaired neuronal synaptic plasticity and upregulated mitophagy levels. (**A**) SH-SY5Y cells were treated with different concentrations of CoCl_2_ (50, 100, 200, 400, 600, 800 μM) for 24 h and CCK-8 was performed to detect the cell viability. (**B**,**C**) The Western blotting of Bax, Bcl-2, caspase-3, and cleaved caspase-3, as well as quantitative analysis. (**D**,**E**) TUNEL staining was performed after treatment with CoCl_2_ (400 μM) for 24 h to validate and quantify the cell death ratio (Scale bar = 50 μm). (**F**,**G**) The Western blotting of PSD-95 and synaptophysin, as well as quantitative analysis. (**H**,**I**) The Western blotting of Drp1, Tom20, PINK1, parkin, LC3, and p62, as well as quantitative analysis. Data are presented as means ± SEM. * *p* < 0.05, ** *p* < 0.01, *** *p* < 0.001, **** *p* < 0.0001 vs. Con, N = 4 for each group.

**Figure 2 cells-13-00872-f002:**
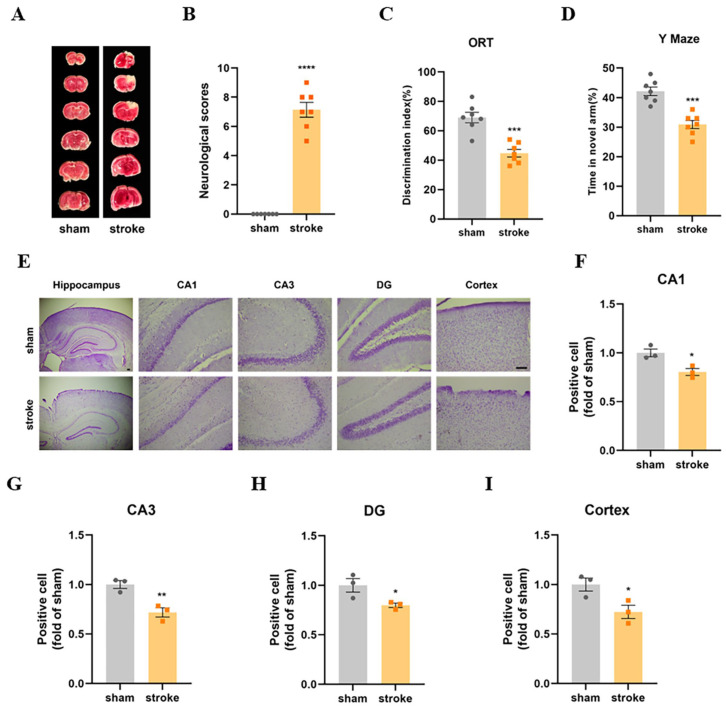
The stroke mice showed cognitive dysfunction and neurologic damage. (**A**) TTC staining of brain sections (N = 4). (**B**) Neurological scores in mice (N = 7). (**C**) The discrimination index in the NOR test (N = 7). (**D**) Time spent in the novel arm of the Y maze (N = 7). (**E**) Nissl staining was used to measure the number of neurons in the hippocampus and cortex regions. (**F**–**I**) The Nissl-positive cells in the hippocampus were quantified by ImageJ. Scale bar = 50 μm (N = 3). Data are presented as means ± SEM. * *p* < 0.05, ** *p* < 0.01, *** *p* < 0.001, **** *p* < 0.0001 vs. Sham.

**Figure 3 cells-13-00872-f003:**
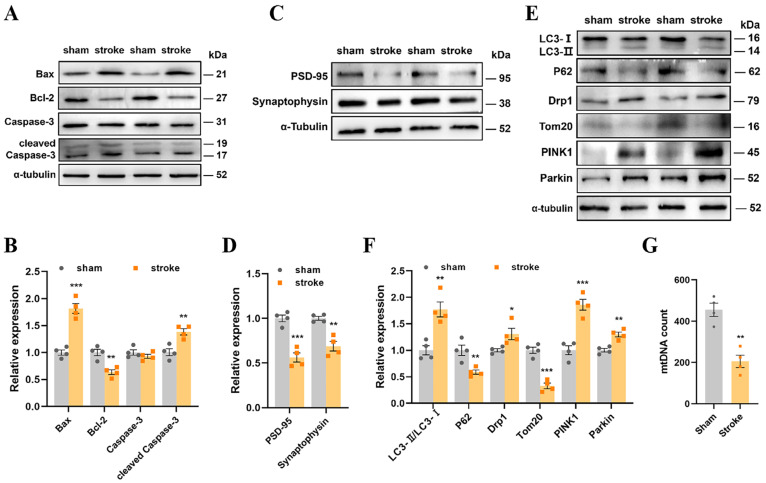
The stroke mice had upregulated mitophagy levels and impaired synaptic plasticity. (**A**,**B**) The Western blotting of Bax, Bcl-2, caspase-3, and cleaved caspase-3, as well as quantitative analysis. (**C**,**D**) The Western blotting of PSD-95 and synaptophysin, as well as quantitative analysis. (**E**,**F**) The Western blotting of Drp1, Tom20, PINK1, parkin, LC3, and p62, as well as quantitative analysis. (**G**) The relative mtDNA copy number detected by qPCR. Data are presented as means ± SEM. * *p* < 0.05, ** *p* < 0.01, *** *p* < 0.001 vs. Sham. N = 4 for each group.

**Figure 4 cells-13-00872-f004:**
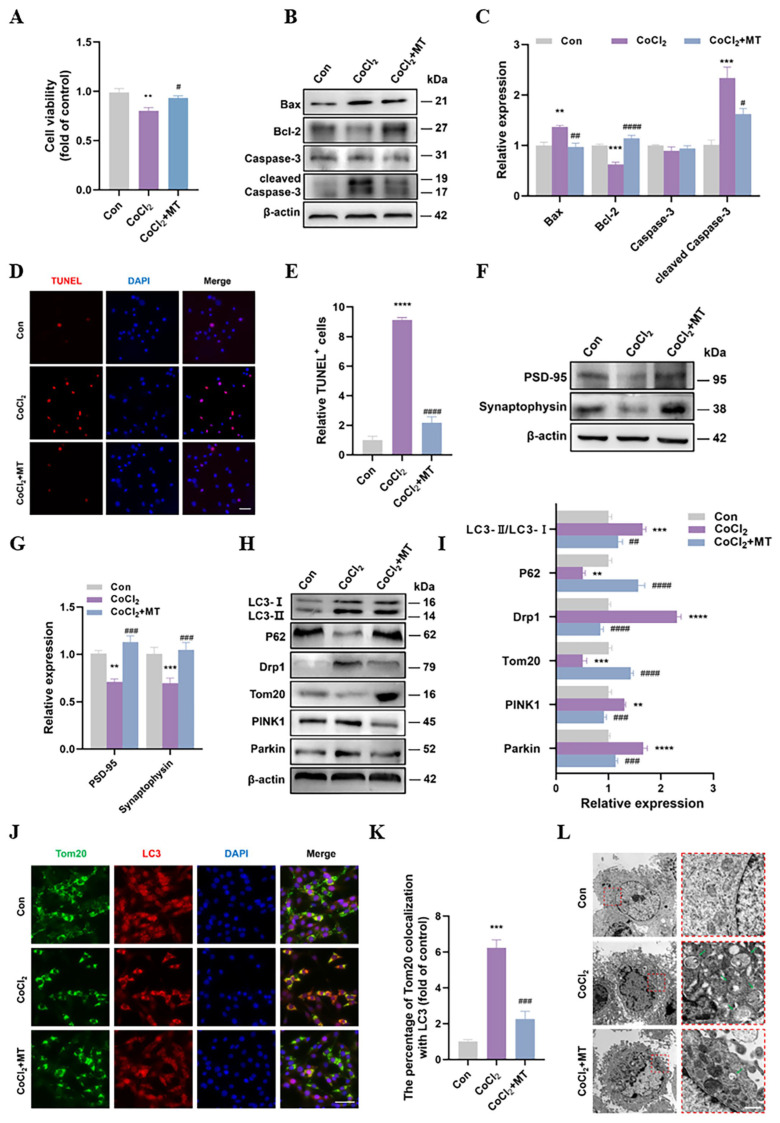
MT protected neuronal synaptic plasticity and downregulated mitophagy levels in vitro. (**A**) SH-SY5Y cells were treated with CoCl_2_ (400 μM) or CoCl_2_ (400 μM) + MT (100 μM) for 24 h and CCK-8 was performed to detect the cell viability. (**B**,**C**) The Western blotting of Bax, Bcl-2, caspase-3, and cleaved caspase-3, as well as quantitative analysis. (**D**,**E**) TUNEL staining was performed to validate the cell death ratio (Scale bar = 50 μm) and quantified. (**F**,**G**) The Western blotting of PSD-95 and synaptophysin, as well as quantitative analysis. (**H**,**I**) The Western blotting of Drp1, Tom20, PINK1, parkin, LC3, and p62, as well as quantitative analysis. (**J**,**K**) Immunofluorescence staining was used to detect the colocalization of Tom20 and LC3, as well as quantitative analysis. Scale bar = 50 µm. (**L**) Mitochondrial structures and mitophagy were observed by TEM. The boxed regions in the left panels are enlarged in the right panels. Scale bar = 500 nm. Green arrow: mitophagy-like event. Data are presented as means ± SEM. ** *p* < 0.01, *** *p* < 0.001, **** *p* < 0.0001 vs. Con, # *p* < 0.05, ## *p* < 0.01, ### *p* < 0.001, #### *p* < 0.0001 vs. CoCl_2_ N = 4 for each group.

**Figure 5 cells-13-00872-f005:**
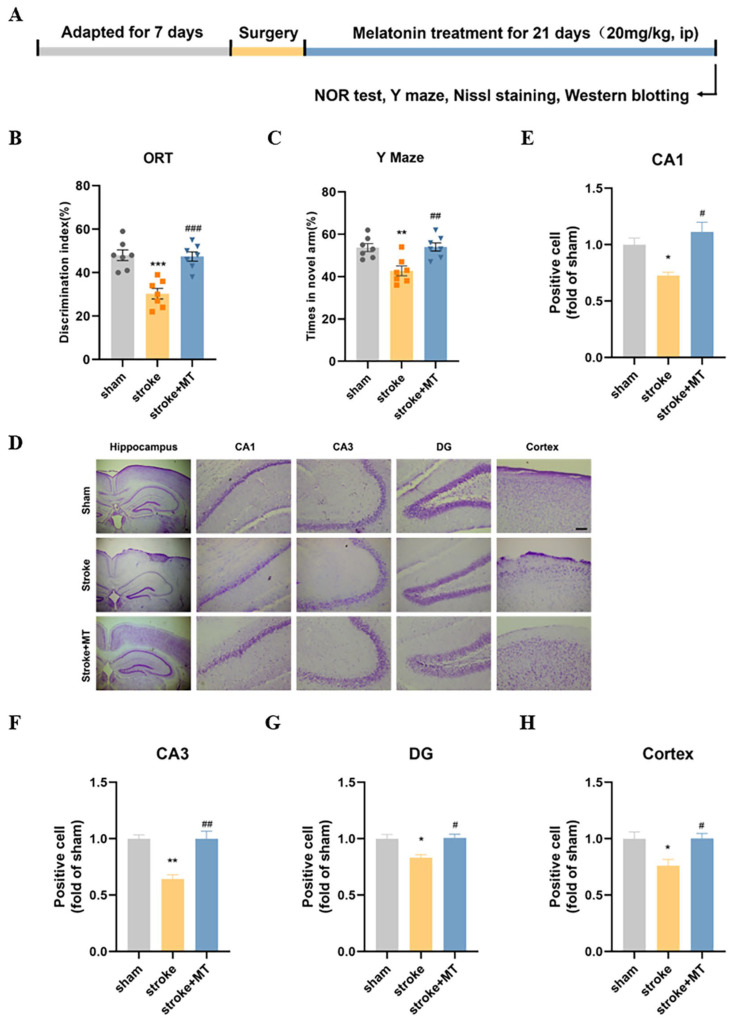
MT ameliorated cognitive defects and reversed neurons loss in stroke mice. (**A**) Experiment procedure. (**B**) The discrimination index in the NOR test (N = 7). (**C**) Time spent in the novel arm of the Y maze (N = 7). (**D**) Nissl staining was used to measure the number of neurons in the hippocampus and cortex regions (**E**–**H**) The Nissl-positive cells in the hippocampus were quantified by ImageJ. Scale bar = 50 μm (N = 3). Data are presented as means ± SEM. * *p* < 0.05, ** *p* < 0.01, *** *p* < 0.001 vs. Con, # *p* < 0.05, ## *p* < 0.01, ### *p* < 0.001 vs. Stroke.

**Figure 6 cells-13-00872-f006:**
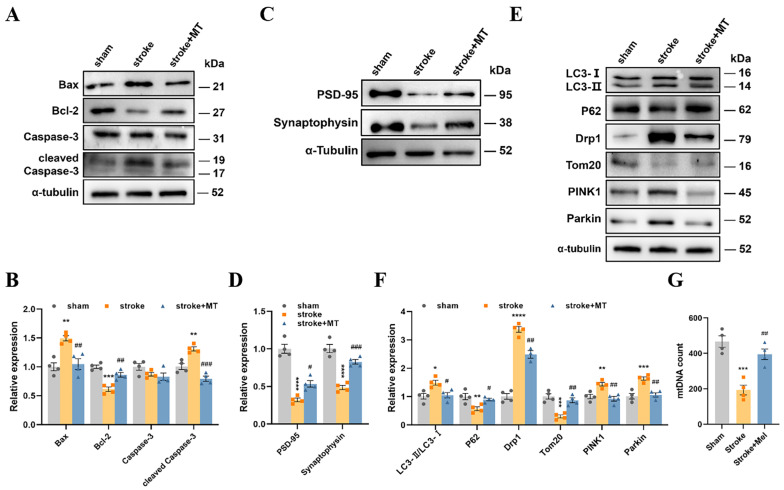
MT ameliorated synaptic plasticity impairments and downregulated mitophagy levels in stroke mice. (**A**,**B**) The Western blotting of Bax, Bcl-2, caspase-3, and cleaved caspase-3, as well as quantitative analysis. (**C**,**D**) The Western blotting of PSD-95 and synaptophysin, as well as quantitative analysis. (**E**,**F**) The Western blotting of Drp1, Tom20, PINK1, parkin, LC3, and p62, as well as quantitative analysis. (**G**) The relative mtDNA copy number detected by qPCR. Data are presented as means ± SEM. * *p* < 0.05, ** *p* < 0.01, *** *p* < 0.001, **** *p* < 0.0001 vs. Con, # *p* < 0.05, ## *p* < 0.01, ### *p* < 0.001 vs. Stroke N = 4 for each group.

## Data Availability

The raw data supporting the conclusions of this article will be made available by the corresponding authors, without undue reservation.

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
