# Peer review of "Melatonin Ameliorates Post-Stroke Cognitive Impairment in Mice by Inhibiting Excessive Mitophagy"

_cells, 2024, doi:10.3390/cells13100872_

Round 1
Reviewer 1 Report
Comments and Suggestions for Authors
Authors have submitted a really well-written manuscript with novel data. To further improve the quality of the report, here are my suggestions:
1) Please provide a more detailed info in the Animals section, i.e., how many animals were housed per caged, info on environmental enrichment. Also, authors state that animals had free access to food and water during 1 week of acclimation. What happened after 1 week? Were the food and water rationed?
2) Were any of the animals excluded after the induction of PT? If so, how many?
3) Please also mention in the TTC section that 4 animals were used (it is found in the legend for Fig. only).
4) Authors could provide the Nissl staining data (Fig. 2F-I) as columns and individual values as scatter, similar to Fig. 2B-D?
Author Response
Dear reviewer:
Thank you for the positive feedback and insightful constructive criticism of our manuscript entitled "Melatonin ameliorates post-stroke cognitive impairment in mice by inhibiting excessive mitophagy". The manuscript has been carefully revised, and point-by-point responses are listed below. We hope that this revision is acceptable, and your favorable consideration of our manuscript is greatly appreciated. The revised sections have been highlighted in yellow.
Comment 1. Please provide a more detailed info in the Animals section, i.e., how many animals were housed per caged, info on environmental enrichment. Also, authors state that animals had free access to food and water during 1 week of acclimation. What happened after 1 week? Were the food and water rationed?
Response: We sincerely thank you for your careful reading. The mice were randomly housed in cages with 5 mice per cage. We housed the mice at a constant temperature (22 ± 2°C) and humidity (50 ± 10%) and provided adequate water and food. In addition, we took all measures to reduce mice numbers and suffering. After a 1-week acclimatization period, the animals still had free access to food and water. We have added these instructions in the “Animals” section according to your comment (highlighted in yellow, Page 3, Line 98).
Comment 2. Were any of the animals excluded after the induction of PT? If so, how many?
Response: Thank you for your comments. In this study, all animals survived after PT modeling and no animals were excluded.
Comment 3. Please also mention in the TTC section that 4 animals were used (it is found in the legend for Fig. only).
Response: We are very sorry for our negligence. In our resubmission of the manuscript, we have added the number of experimental animals in the “TTC staining” section. Thank you for your correction (highlighted in yellow, Page 3, Line 128).
Comment 4. Authors could provide the Nissl staining data (Fig. 2F-I) as columns and individual values as scatter, similar to Fig. 2B-D?
Response: Thank you for your comments. Following your suggestion, we have added data points to the bar plot of the Nissl staining (Page 7, Figure 2F-I).
Once again, thank you very much for your comments and suggestions.
Reviewer 2 Report
Comments and Suggestions for Authors
In this work, the authors have demonstrated that melatonin can inhibit excessive mitophagy in-vitro as well as in-vivo in ischemic stroke models. Additionally, they have shown that melatonin can mitigate cognitive deficits and neuronal loss. The findings are interesting but there is lack of a clear mechanistic link between inhibition of mitophagy by melatonin and downregulation of apoptosis. One major concern is that western blots for Figure 4 (in-vitro) and Figure 6 (in-vivo) are the same. The authors should correct that. My specific comments are below:
1) The authors should examine mitophagy in a more reliable and conventional way like mito-Keima or mitoQC.
2) The authors show that mitophagy related proteins are upregulated in ischemic stroke models. Although western blot for Tom20 show significant downregulation, the authors should also verify if mitochondrial content is reduced and whether melatonin prevents reduction of mtDNA. It is also advisable to investigate mitochondrial membrane potential.
3) Can inhibiting mitophagy (by silencing Pink/Parkin) have similar effects as melatonin in suppressing apoptosis?
4) Fig 2E-H: Please show the data points for the 3 animals that were used in the bar plot.
5) For all the data involving animal studies, please show each data-point, representing each animal, in the bar plot ( eg. Fig. 3, 6 etc)
6) Figure 4J: Image resolution is very low, as a result, the co-localization of Tom20 and LC3 is not obvious. Please use a higher mag image and also the data needs to be quantified.
The authors show in their western blot that Tom20 level is significantly downregulated with CoCl2 treatment, but this is not reflected in their fluorescence image. If the exposure or brightness/contrast of Tom20 has been set differently for each sample to show co-localization with LC3, it should be clearly stated in the figure legend.
7) Fig 6: The western blots for mitophagy proteins and a-tub are the same as shown in figure 4. Additionally, the animal to animal variability seems very low in the western blots, which can be possible but nonetheless, it is desirable that the authors show atleast 2 animals, similar to their data in figure 1 and 3.
Author Response
Dear reviewer:
Thank you for the positive feedback and insightful constructive criticism of our manuscript entitled "Melatonin ameliorates post-stroke cognitive impairment in mice by inhibiting excessive mitophagy". The manuscript has been carefully revised, and point-by-point responses are listed below. We hope that this revision is acceptable, and your favorable consideration of our manuscript is greatly appreciated. The revised sections have been highlighted in yellow.
Comment 1. The authors should examine mitophagy in a more reliable and conventional way like mito-Keima or mitoQC.
Response: Thank you for this valuable suggestion. We totally agree with your comment. Indeed, mito-Keima or mitoQC is a more reliable and conventional way to detect mitophagy. However, we believe that the present experimental methods can still accurately detect mitophagy. We reviewed a large amount of literature and concluded that Tom20 (Translocase of outer mitochondrial membrane 20) is a specific mitochondrial marker, and that mitophagy can be accurately examined by double immunofluorescence staining with Tom20 and LC3 (autophagy marker). As in this study, articles by Wu et al. (doi:10.1016/j.phymed.2021.153884), Mao et al. (doi:10.1016/j.phymed.2022.154111), Zhang et al. (doi:10.1111/jpi.12542), and Cai et al. (doi:10.1016/j.redox.2020.101792), all used this method to detect mitophagy.
Comment 2. The authors show that mitophagy related proteins are upregulated in ischemic stroke models. Although western blot for Tom20 show significant downregulation, the authors should also verify if mitochondrial content is reduced and whether melatonin prevents reduction of mtDNA. It is also advisable to investigate mitochondrial membrane potential.
Response: We sincerely thank you for your careful reading. Tom20 is located in the outer mitochondrial membrane and has been widely applied as a mitochondrial marker protein, and its level can accurately reflect the mitochondrial content. In addition, following your suggestion, we examined the content of mtDNA. The results showed that mtDNA content was significantly reduced in stroke mice (Page 8, Figure 3G), and melatonin could prevent the reduction of mtDNA (Page 11, Figure 6G). We agree with your comments that the test on mitochondrial membrane would be useful to support our conclusion, however, after careful consideration of the funding and the deadline of the revision, we found it is impractical to implement the related experiments. However, we believe that the present results can still support the conclusion of this paper. The aim of this paper is to explore whether mitophagy plays a role in the improvement of PSCI by melatonin. We examined the levels of mitophagy in vivo and in vitro by using various methods such as western blot, immunofluorescence, and transmission electron microscopy, all of which supported our conclusions. In the future study, we will perform related studies for deeply and thoroughly understand this problem. Thank you very much for your insightful constructive idea.
Comment 3. Can inhibiting mitophagy (by silencing Pink/Parkin) have similar effects as melatonin in suppressing apoptosis?
Response: Thank you for your comments. Zeng et al. showed that pretreatment with 3-MA (an autophagy inhibitor) could inhibit excessive mitophagy and suppressed apoptosis in PC12 and HT22 cells, which had the similar effects as melatonin (doi:10.1007/s10753-021-01579-9). This article is also mentioned in the “discussion” section of this article (highlighted in yellow, Page 13, Line 445). However, whether silencing Pink/Parkin has a similar effect as melatonin in inhibiting apoptosis has not been directly demonstrated and needs to be further investigated.
Comment 4. Fig 2E-H: Please show the data points for the 3 animals that were used in the bar plot.
Response: Thank you for your comments. Following your suggestion, we have added data points in the bar plot of the Nissl staining (Page 7, Figure 2F-I).
Comment 5. For all the data involving animal studies, please show each data-point, representing each animal, in the bar plot (eg. Fig. 3, 6 etc)
Response: Thank you for your comments. Following your suggestion, for all the data involving animal studies, we have added each data-point representing each animal in the bar plot.
Comment 6. Figure 4J: Image resolution is very low, as a result, the co-localization of Tom20 and LC3 is not obvious. Please use a higher mag image and also the data needs to be quantified.
The authors show in their western blot that Tom20 level is significantly downregulated with CoCl2 treatment, but this is not reflected in their fluorescence image. If the exposure or brightness/contrast of Tom20 has been set differently for each sample to show co-localization with LC3, it should be clearly stated in the figure legend.
Response: We sincerely thank you for your careful reading. Following your suggestion, we have replaced the Figure 4 with a higher resolution image and quantified Figure 4J (Page 9, Figure 4). To avoid misunderstandings, we have modified the images in Figure 4J and used more typical images. The exposure and brightness of all images are consistent (Page 9, Figure 4J). It is also worth mentioning that the cell density is decreased after CoCl2 treatment, so the change in Tom20 in Western blotting will be more significant.
Comment 7. Fig 6: The western blots for mitophagy proteins and a-tub are the same as shown in figure 4. Additionally, the animal to animal variability seems very low in the western blots, which can be possible but nonetheless, it is desirable that the authors show at least 2 animals, similar to their data in figure 1 and 3.
Response: We deeply apologize for our carelessness. In our resubmission of the manuscript, we have replaced it with the correct western blots (Page 9, Figure 4H). Thank you for your correction. We apologize for not providing enough western blots with 2 animals on the same membrane. As a supplement, we have added data points representing each animal in the bar graph.
Once again, thank you very much for your comments and suggestions.
Round 2
Reviewer 2 Report
Comments and Suggestions for Authors
The authors have addressed all my concerns.